# Modified Mini-Keystone Flaps for Coverage of Tiny Volar Pulp Defects of the Fingertips in Cases with Missing Amputation Skin Stumps: A Retrospective Study

**DOI:** 10.3390/jcm11123394

**Published:** 2022-06-13

**Authors:** Byung Woo Yoo, Seungyoon Oh, Junekyu Kim, Kap Sung Oh, Hyun Woo Shin, Kyu Nam Kim

**Affiliations:** Department of Plastic and Reconstructive Surgery, Kangbuk Samsung Hospital, Sungkyunkwan University School of Medicine, 29, Saemunan-ro, Jongno-gu, Seoul 03181, Korea; dbansdj@naver.com (B.W.Y.); yoona1507@gmail.com (S.O.); kokoro72@naver.com (J.K.); kapsung.oh@samsung.com (K.S.O.); mdshin7@naver.com (H.W.S.)

**Keywords:** fingertip defect, tiny defect, volar pulp defect, keystone flap, flap coverage

## Abstract

This study aimed to demonstrate the expanding versatility of keystone flap reconstruction in fingertips. Fifteen patients who underwent the modified mini-keystone flap reconstruction for tiny volar pulp defects of the fingertip between September 2020 and February 2021 were included in this study (average age: 43.4 ± 13.52 years, range: 19–61 years). Patient data were retrospectively collected from their medical records. The two-point discrimination test was used to evaluate the degree of sensory recovery. All defects were successfully covered with the modified mini-keystone flap. The defect sizes ranged from 0.5 cm × 1 cm to 1.2 cm × 2.0 cm, and the flap sizes ranged from 0.7 cm × 1.5 cm to 1.5 cm × 3.0 cm. Although one patient showed a small distal margin maceration, all flaps survived fully. The overall outcomes were favorable at the mean follow-up period of 5.73 ± 0.79 months. We suggest that the modified mini-keystone flap technique is a promising alternative modality for covering tiny volar pulp defects of the fingertip, with few complications and favorable outcomes.

## 1. Introduction

Fingertip defects are common injuries of the upper extremities because the fingertip is the most peripheral part of the body. However, treatment of these injuries is challenging because of the prominent role of the fingertips in an individual’s activities [1]. In particular, tiny volar pulp defects of the fingertips are a common problem in daily life. Tiny defects can be defined as very small wounds for which primary closure is difficult to achieve due to the loss of some skin and soft tissues and reduction in elasticity in their location. These defects are often caused by trauma, such as cuts with a knife, saw, or sharp edge. In cases with an existing amputation skin stump, a skin graft using the stump is the best option for coverage. However, when amputation skin stumps are unavailable, several methods, such as conservative dressing, skin grafts from other donor sites, and local flaps, can be used. Each of these techniques have both advantages and disadvantages. In the absence of an ideal method, a particular defect can be managed using various options to yield similar results [2,3]. Generally, superficial defects can be covered with skin grafts, and flap techniques are useful to cover deep defects with exposure of the underlying structures [3]. Surgeons usually choose an appropriate reconstructive method on a case-by-case basis according to the location and size of each defect [4]. Although various flap techniques have been devised to cover fingertip defects, new or modified techniques can yield favorable or better outcomes compared with previous methods. Herein, we present a retrospective review of our clinical experience with tiny volar pulp defect coverage using a modified mini-keystone flap (m-KF), that is, a combination of omega variation (OV) closure and Sydney Melanoma Unit Modification (SMUM). Through this study, we demonstrate the expanding versatility of KF reconstruction in fingertips.

## 2. Materials and Methods

This study was approved by the Institutional Review Board of our hospital. All research procedures in this study were performed in accordance with the guidelines proposed in the 1975 Declaration of Helsinki. All the patients in this study provided written consent to publish their information and images in an online open access publication before undergoing the procedures and operations.

In this study, we included patients who underwent modified m-KF reconstruction for tiny volar pulp defects of the fingertip between September 2020 and February 2021. Patients who underwent fingertip defect reconstruction using other methods, such as skin grafts and other flaps, were excluded. Patients’ data from medical records and clinical photographs were retrospectively reviewed to obtain information regarding defect causes, defect locations, defect sizes, flap sizes, flap survival, complications, two-point discrimination test (2-PDT) results, and the findings of follow-up assessments for each patient. We routinely obtained 2-PDT measurements to evaluate the degree of sensory recovery at the final postoperative follow-up of each patient.

### 2.1. Surgical Techniques

After debridement of the lesion of the fingertip, the final defect was identified, and modified m-KF reconstruction was performed. This flap was named m-KF because it was much smaller than the ubiquitous KF. We used OV closure [5] and SMUM [6] KF in all cases. The OV KF is a modification of the conventional KF in which the original defect is closed in a fish-mouth fashion through additional rotational movement [4,5,7]. The SMUM KF entails maintaining a skin bridge along the greater arch of the KF [4,6,7]. The modified m-KF was designed to be slightly larger than the defect at the side of sufficient tissue laxity. A skin incision was made along the flap with the remaining skin bridge, and dissection progressed to the subcutaneous tissues. The flap margin was minimally undermined to achieve OV closure. Fastidious hemostasis was achieved, and flap inset was performed in the following sequence: a three-point area (tip of a fish-mouth closure) of the OV was sutured first, both ends of the V-Y closure were sutured next, and the donor side (greater arc) of the m-KF was sutured at the end. A simple dressing with foam material was applied at the end of the operation. Figure 1 presents the schematic diagrams of the modified m-KF reconstruction for tiny volar pulp defects of the fingertip.

### 2.2. Evaluation of Postoperative Functional Outcomes

We recorded the degree of sensory recovery at the final postoperative follow-up examination by using the Touch Test^®^ 2-Point Discriminator (North Coast Medical Inc., Morgan Hill, CA, USA) to measure static and dynamic two-point discrimination. All tests were performed and recorded for the flap coverage area of the affected fingertip and the equivalent area of the contralateral normal fingertip by our senior author. For the static two-point discrimination test (2-PDT), values of <6 mm, 6–10 mm, and >11 mm indicated good, fair, and poor discrimination, respectively [8,9,10]. For the dynamic 2-PDT, values of <4 mm, 4–7 mm, and >8 mm indicated good, fair, and poor discrimination, respectively [8,9,10]. Figure 2 shows the two-point discriminator and the 2-PDT setup used in this study.

### 2.3. Statistical Analysis

We used GraphPad Prism version 8.4.3 (GraphPad Software Inc., San Diego, CA, USA) software for all statistical analyses. Continuous variables were expressed as mean ± standard deviation (SD). We used Student’s *t*-test for continuous variables to compare the differences between 2-PDT at the affected fingertip and that at the contralateral normal fingertip. The significance level was set at *p* < 0.05.

## 3. Results

The patients’ clinical data are summarized in Table 1. A total of 15 patients (10 male and 5 female patients) aged 19–61 years (average age, 43.4 ± 13.52 years) were included in this study. The defects were caused by traumatic injury in all cases. All defects corresponded to the aforementioned definition of a tiny full-thickness defect with no bone exposure. The sizes of the defects ranged from 0.5 cm × 1.0 cm to 1.2 cm × 2.0 cm. Successful coverage of all defects was achieved by using m-KF combined with OV and SMUM. The flap sizes ranged from 0.7 cm × 1.5 cm to 1.5 cm × 3.0 cm. All flaps survived without any flap-related complications, such as venous congestion and arterial insufficiency. One patient had small-sized macerations of the distal margin; however, this was resolved with conservative treatment and required no further surgical management. All other patients were completely healed without any wound complications at 2-week postoperative follow-up. At the average follow-up period of 5.73 ± 0.79 months (range, 5–7 months), the mean static 2-PDT value at the affected and the contralateral normal fingertips were 4.00 ± 1.13 mm and 3.60 ± 0.63 mm, respectively (*p* = 0.243). The mean dynamic 2-PDT value at the affected and contralateral normal fingertips were 3.27 ± 1.03 mm and 2.87 ± 0.64 mm, respectively (*p* = 0.213). All 2-PDT values were above fair, and all patients were adequately satisfied with their final outcomes. Table 2 summarizes the 2-PDT data. In the following section (Case Presentations), we describe some representative cases to elucidate our m-KF reconstruction method for tiny volar pulp defects in fingertips.

### 3.1. Case Presentations

#### 3.1.1. Case 5 

A 47-year-old man sustained a right second fingertip injury while using a cutter knife. The patient’s amputation skin stump was not available. Under a digital nerve block, we performed debridement and identified a 0.6 cm × 1.0 cm sized tiny volar pulp defect (Figure 3). We covered the defect with a modified m-KF (0.8 cm × 1.5 cm) from the proximal side of the defect. Flap insertion and donor-site closure with minimal tension were achieved. The flap fully survived without any postoperative complications. At the 6-month follow-up, the static 2-PDT value was 2 mm in the affected fingertip and 2 mm in the contralateral normal fingertip, and the dynamic 2-PDT value was 2 mm in the affected fingertip and 2 mm in the contralateral normal fingertip. The patient was satisfied with the final outcome.

#### 3.1.2. Case 6 

A 36-year-old man sustained a left fifth fingertip injury while using a cutter knife. The patient’s amputation skin stump was unavailable. Under a digital nerve block, we performed debridement and identified a 0.6 cm × 1.5 cm sized tiny volar pulp defect (Figure 4). The defect was covered with a modified m-KF (0.8 cm × 2.5 cm) from the proximal side of the defect. Flap insertion and donor-site closure with minimal tension were achieved. The flap survived without any postoperative complications. At the 6-month follow-up, the static 2-PDT value was 4 mm in both the affected fingertip and the contralateral normal fingertip, while the dynamic 2-PDT value was 3 mm in both the affected and contralateral normal fingertips. The patient was satisfied with the final outcome.

#### 3.1.3. Case 11 

A 30-year-old man sustained a left thumb fingertip injury while using a kitchen knife. The patient’s amputation skin stump was unavailable. Under a digital nerve block, we performed debridement and identified a 1.2 cm × 2.0 cm sized tiny volar pulp defect (Figure 5). We covered the defect using a modified m-KF (1.5 cm × 3.0 cm) from the proximal side of the defect. Flap insertion and donor-site closure with minimal tension were achieved. The flap survived without any postoperative complications. At the 5-month follow-up, the static 2-PDT value was 6 mm in the affected fingertip and 4 mm in the contralateral normal fingertip, and the dynamic 2-PDT value was 5 mm in the affected fingertip and 3 mm in the contralateral normal fingertip. The patient was satisfied with the final outcome.

## 4. Discussion

This retrospective study reports our successful experience with modified m-KF reconstruction for tiny volar pulp defects of the fingertips. As mentioned earlier, a wide variety of reconstructive options have been developed for fingertip defects, with scarce evidence supporting one method over the other [2]. Therefore, the ideal treatment for different types of fingertip injuries remains debatable [1,2,11,12]. Generally, the fundamental reconstructive principles include replacing like-with-like tissues, minimizing donor-site morbidity, and recovering normal functional properties. Considering these objectives, free flaps are suitable for large fingertip reconstruction, whereas local flaps can be the best choice for small-to-moderate-sized defect coverage of the fingertip [1,2,11]. A variety of local flaps can be used to cover fingertip defects [1,2,11]. V-Y advancement flaps, rotation flaps, transposition flaps, and bilobed flaps can be used to cover small-sized fingertip defects [1,2,11,13]. However, homo-digital and hetero-digital island flaps, cross-finger flaps, and dorsal metacarpal artery perforator flaps are required to cover moderate-sized defects in the fingers [2,14,15].

KF, devised by Behan in 2003, is one of the most meaningful innovations in the field of reconstructive surgery over the last 20 years [3,4,7,16]. KF is characterized by its simple and defect-adaptive design, with a curvilinear-shaped trapezoidal design consisting of two conjoined V-Y flaps moving in the horizontal direction on the longitudinal flap axis [3,4,7,16]. A large number of studies have shown that KF reconstruction can cover various anatomical defects in the human body, including the head, neck, trunk, and extremities [4,7,16,17,18]. In contrast, only a few reports have described KF finger reconstruction [19,20]. Although no previous article has described KF fingertip reconstruction, a literature review using PubMed and Google Scholar with search terms including “keystone flap” AND “finger” identified two articles describing KF finger reconstruction, which were case reports [19,20]. To the best of our knowledge, our study presents the first case series of KF reconstruction for fingertip defects.

The original classification of KF by Behan covered four subtypes: type I (skin incision only), type II (A, division of the deep fascia along the outer curvilinear line; B, division of the deep fascia and skin graft to the secondary defect), type III (opposing keystone flaps designed to create a double-keystone flap), and type IV (keystone flap with undermining of up to 50% of the flap subfascially) [16]. In addition to these original subtypes, two representative modifications have been devised: the OV KF and the SMUM KF [4,5,6,7]. As mentioned earlier, we used a combination of OV and SMUM KFs in all cases of the present study. The OV KF can provide further flap movement via additional rotational movement as mentioned above, and, consequently, it can reduce tension as well as avoid sacrifice of healthy tissues during wound closure [4,5,7]. SMUM KF can allow additional vascularity, preserve the subdermal lymphatics, and provide structural stabilization of the flap through the skin bridge [4,6,7]. When small-sized flaps for covering tiny defects are applied as m-KFs, these two modifications are very helpful in acquiring sufficient flap movement and maintaining hemodynamic flap stability while performing minimal undermining of the flap. Through the combination of these two modifications, we achieved complete flap survival with no significant postoperative complications in all of our cases.

In tiny volar pulp defects of the fingertip, skin grafts using the amputation skin stump are generally used as a simple and optimal option; however, this option is frequently not available due to the absence of the skin stump or because of contaminated amputation [1]. In such cases, three treatment methods, namely, secondary intention with conservative dressing, skin graft from another donor site, and local flaps, can be considered. Secondary intention with conservative dressing, such as occlusive or semi-occlusive dressing, may actually represent the simplest solution for any small and superficial fingertip defects that do not involve the bone. However, it is usually a time-consuming treatment method and is often compromised by delayed wound healing due to an underlying disease or concurrent wound infection. Furthermore, patients may complain of persistent pain until epithelialization progresses to some extent. Through fingertip defect coverage with skin graft and local flap, surgeons can perform more time-saving treatment methods and achieve reliable wound healing for patients. Skin grafts are commonly used to cover defects of various sizes without exposure of vital structures in reconstructive surgery [3,7]. However, they have some limitations, including vulnerability to mechanical forces, decreased sensation, color and texture mismatch, and donor-site morbidity [3,7]. Reconstruction with the local flap technique is superior to other options because it allows reconstruction of the volar pulp in the fingertip with a tissue showing similar features [1,21]. The volar pulp skin of the fingertip comes under a lot of weight during hand function because of its specialized sensation and grasp abilities [1]. Thus, the ideal reconstruction outcomes of the volar pulp area can be achieved by coverage with local tissues that have similar skin durability while preserving sensation [1,21]. In the present study, our modified m-KF technique for covering tiny volar defects of the fingertip yielded favorable outcomes in all cases. In particular, the functional outcomes of our study were remarkable in terms of sensation recovery. The mean values of both the static and dynamic 2-PDT at the affected fingertip did not show a significant difference from the corresponding values at the contralateral normal fingertip in the present study (*p* = 0.243 and *p* = 0.213) (Figure 6). Furthermore, all 2-PDT values were more than fair in all cases. On the basis of these results, we inferred that the modified m-KF technique can provide outstanding sensation recovery in tiny volar pulp reconstruction.

Because fingers perform various movements and lack sufficient surrounding tissues, securing minimal wound tension when covering finger defects is very important [19]. An attempt to close the defect under tension can result in a high risk of wound-related complications [19]. Although the volar pulp area of the fingertip has a relatively consistent amount of tissue compared with that of other finger regions, closing the wound without tension is still difficult, and wound closure with minimal tension is obviously essential. Recruitment and rearrangement of tissue laxity is a crucial biomechanical property of KF reconstruction, which results from the use of two conjoined V-Y advancement flaps at each end of the KF by redistributing the wound tension perpendicular to the line of advancement [4,7,18]. This redistribution of the wound tension results in a tension-reducing effect of KF in wound closure [4,7,18]. In the present study, we considered that the modified m-KF technique also allowed the wound to be closed with minimal tension when covering tiny volar defects despite the limited amount of extra tissue in the fingertip. Thus, there were no significant wound problems requiring surgical intervention or revision in our cases. All surgical procedures and postoperative care in this study were performed in the outpatient clinic, indicating that the modified m-KF technique is simple and convenient.

Despite the successful outcomes, our study had several limitations. This study included a very small number of cases, was a nonrandomized retrospective study, and had no comparison group. Therefore, selection bias and confounding factors were inevitably encountered, which affected the study outcomes. A well-designed prospective large-scale study with other comparison groups is required to confirm the favorable and consistent outcomes of our study. Meanwhile, with regard to the choice of reconstructive surgery, our modified m-KF is not an optimal method, although it can be a reliable alternative for fingertip defects. Reconstructive surgeons do not have to adhere to the use of the modified m-KF for the reconstruction of tiny fingertip defects in all cases. Finally, any scar at the volar pulp of the fingertip can be painful and irritating for patients. Limited aesthetic results, such as change in fingertip contour, can occur due to surgical reconstructions; therefore, surgical procedures such as our modified m-KF should be performed with caution. As mentioned earlier, we designed the flap at the side of sufficient tissue laxity (Figure 1). The abovementioned tension-reducing effect of the modified m-KF can allow wound closure with minimal tension, possibly helping in minimizing scar formation in the volar pulp area. Despite these limitations, our study findings are significant because this study is the first consecutive case series that describes fingertip defect reconstruction using KF conducted by a single surgeon (the corresponding author of this study) and demonstrates the expanding versatility of KF in reconstructive surgery.

## 5. Conclusions

Based on the successful outcomes reported in this study, we believe that the modified m-KF technique can be a good alternative to other reconstructive modalities for covering tiny volar pulp defects of the fingertip in terms of assurance of reliable outcomes (outstanding sensation recovery) and fulfillment of ideal reconstruction (replacement of like-with-like tissue). Furthermore, we plan to apply KF reconstruction to other finger and hand regions to enhance and expand the usefulness of the KF technique based on our study findings.

## Figures and Tables

**Figure 1 jcm-11-03394-f001:**
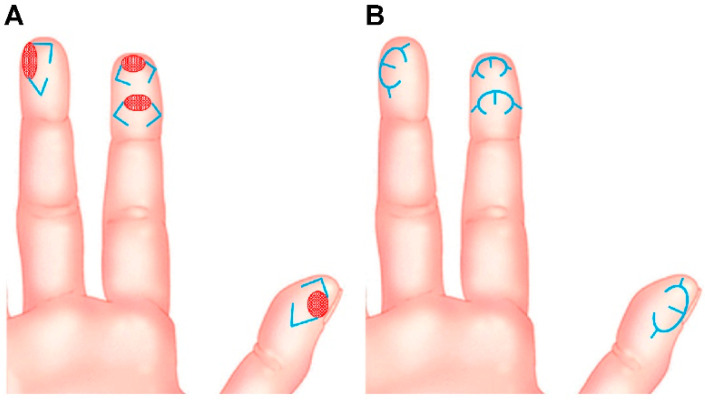
Schematic diagrams of modified mini-keystone flap (m-KF) reconstruction for tiny volar defects of the fingertip. (**A**) Various tiny volar defects of the fingertips (red-colored ovals) and designs of the modified m-KFs (blue-colored lines). (**B**) Final appearance after coverage of the modified m-KFs (blue-colored lines).

**Figure 2 jcm-11-03394-f002:**
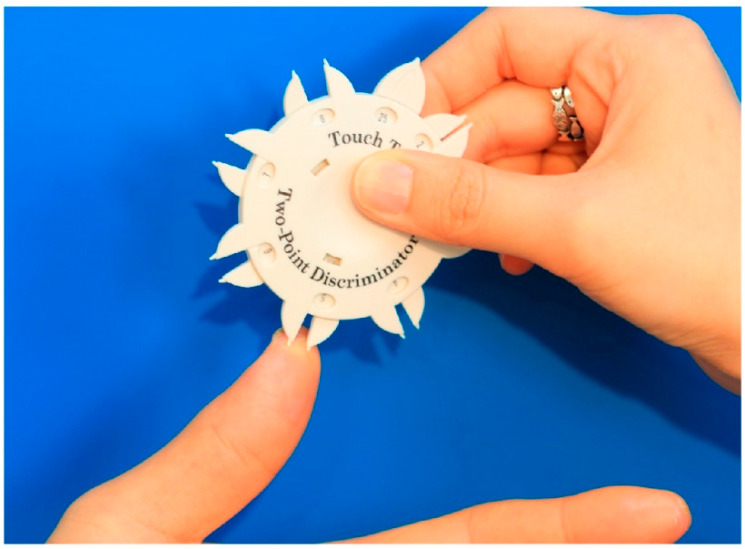
The two-point discrimination test (2-PDT). To perform the 2-PDT, the evaluator touches the patient’s fingertip with the two-point discriminator device, randomly alternating between one and two points. The patient is asked to report whether one or two points were felt.

**Figure 3 jcm-11-03394-f003:**
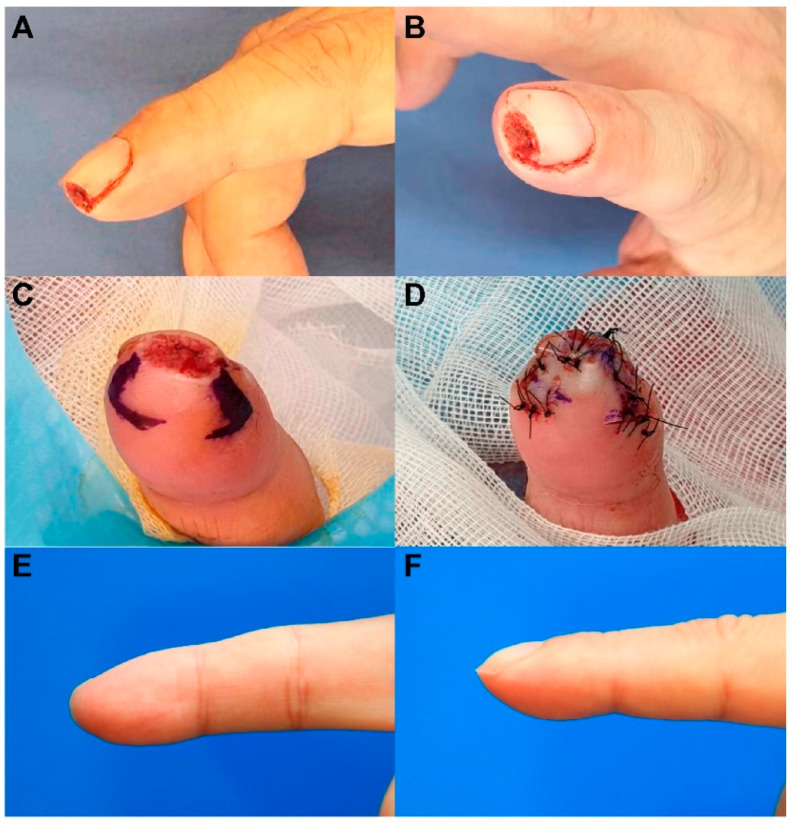
Clinical photographs of a 47-year-old man who sustained a right second fingertip injury while using a cutter knife. (**A**,**B**) A tiny volar pulp defect of the right second fingertip (0.6 cm × 1 cm). (**C**) Design of a modified mini-keystone flap (m-KF) (0.8 cm × 1.5 cm). (**D**) Successful coverage of the defect with the modified m-KF. (**E**,**F**) Postoperative photographs obtained at the 6-month follow-up.

**Figure 4 jcm-11-03394-f004:**
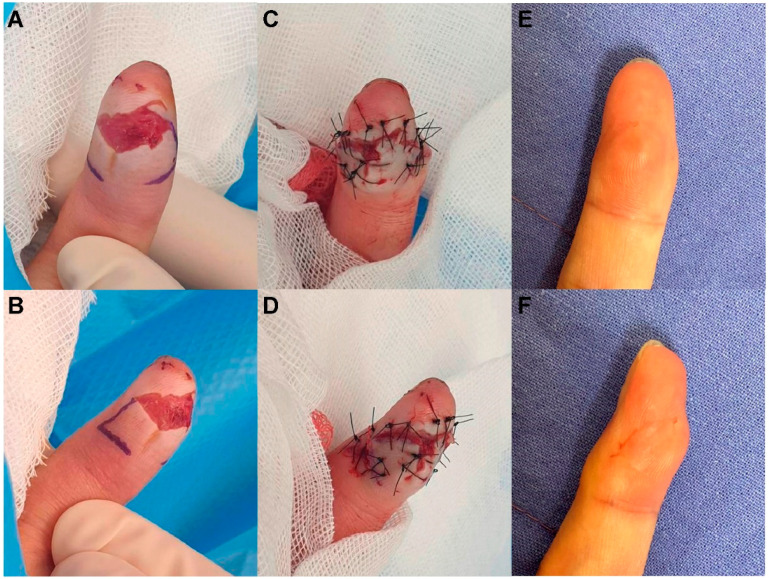
Clinical photographs of a 36-year-old man who sustained a left fifth fingertip injury caused by a cutter knife. (**A**,**B**) A tiny volar pulp defect of the right second fingertip (0.6 cm × 1.5 cm), and design of a modified mini-keystone flap (m-KF) (0.8 × 2.5 cm). (**C**,**D**) Successful coverage of the defect with the modified m-KF. (**E**,**F**) Postoperative photographs obtained at the 6-month follow-up.

**Figure 5 jcm-11-03394-f005:**
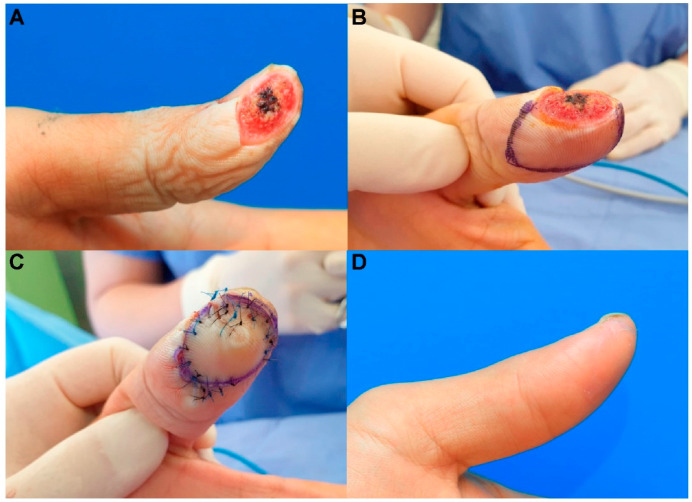
Clinical photographs of a 30-year-old man who sustained a left thumb fingertip injury while using a kitchen knife. (**A**) A tiny volar pulp defect of the left thumb fingertip (1.2 cm × 2.0 cm). (**B**) Design of a modified mini-keystone flap (m-KF) (1.5 cm × 3.0 cm). (**C**) Successful coverage of the defect with the modified m-KF. (**D**) Postoperative photographs obtained at the 5-month follow-up.

**Figure 6 jcm-11-03394-f006:**
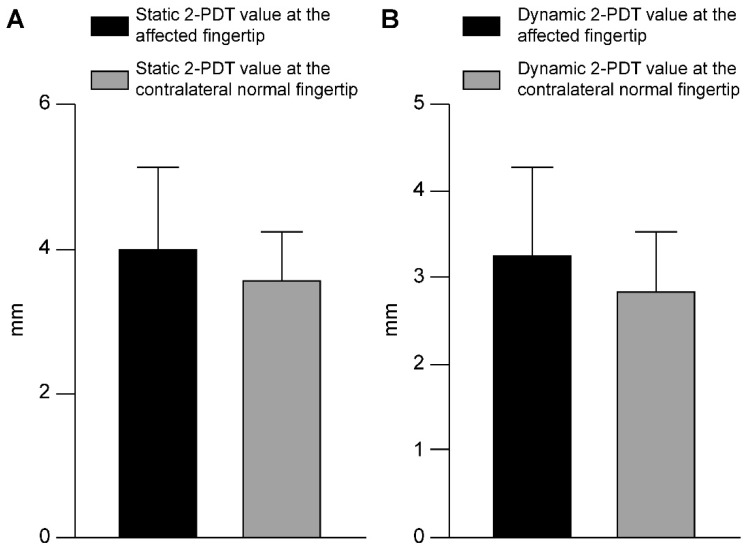
Comparison of mean differences between continuous variables in the two-point discrimination test (2-PDT) data. (**A**) Static 2-PDT. (**B**) Dynamic 2-PDT.

**Table 1 jcm-11-03394-t001:** Patients’ data.

Case	Age/Sex	Cause of the Defect	Defect Location	Defect Size (cm^2^)	Flap Size (cm^2^)	Flap Survival	Complications	Static 2-PDT (Affected Fingertip/Contralateral Normal Fingertip, mm)	Dynamic 2-PDT (Affected Fingertip/Contralateral Normal Fingertip, mm)	Follow-Up Period (Months)
1	F/27	Trauma caused by cutter knife	Rt. volar pulp of the thumb	0.7 × 1.0	0.9 × 2.0	Fully survived	None	3/3	2/2	7
2	F/59	Trauma caused by kitchen knife	Lt. volar pulp of the fifth finger	0.5 × 1.0	0.7 × 1.5	Fully survived	None	4/4	3/3	6
3	M/61	Trauma caused by cutter knife	Rt. volar pulp of the thumb	1.0 × 1.5	1.5 × 3.0	Fully survived	Distal margin maceration	6/4	5/3	5
4	M/34	Trauma caused by cutter knife	Lt. volar pulp of the fourth finger	0.7 × 0.9	0.9 × 1.6	Fully survived	None	3/3	2/2	7
5	M/47	Trauma caused by cutter knife	Rt. volar pulp of the second finger	0.6 × 1.0	0.8 × 1.5	Fully survived	None	2/2	2/2	6
6	M/36	Trauma caused by cutter knife	Lt. volar pulp of the fifth finger	0.6 × 1.5	0.8 × 2.5	Fully survived	None	4/4	3/3	6
7	M/19	Trauma caused by cutter knife	Rt. volar pulp of the thumb	0.8 × 1.1	1.0 × 2.3	Fully survived	None	3/3	2/2	7
8	M/41	Trauma caused by kitchen knife	Lt. volar pulp of the second finger	0.6 × 1.0	0.8 × 2.0	Fully survived	None	4/4	3/3	6
9	M/58	Trauma caused by cutter knife	Rt. volar pulp of the second finger	0.9 × 1.5	1.3 × 2.5	Fully survived	None	4/4	4/4	5
10	M/56	Trauma caused by cutter knife	Lt. volar pulp of the second finger	0.6 × 1.2	0.9 × 1.6	Fully survived	None	5/4	4/3	5
11	M/30	Trauma caused by kitchen knife	Lt. volar pulp of the thumb	1.2 × 2.0	1.5 × 3.0	Fully survived	None	6/4	5/3	5
12	M/31	Trauma caused by kitchen knife	Lt. volar pulp of the second finger	0.7 × 1.2	1.0 × 2.2	Fully survived	None	3/3	3/3	5
13	F/42	Trauma caused by cutter knife	Rt. volar pulp of the fifth finger	0.6 × 1.1	0.8 × 1.7	Fully survived	None	4/4	4/4	5
14	F/56	Trauma caused by kitchen knife	Rt. volar pulp of the fourth finger	0.7 × 1.3	1.1 × 1.8	Fully survived	None	4/4	3/3	5
15	F/54	Trauma caused by kitchen knife	Lt. volar pulp of the second finger	0.8 × 1.0	1.2 × 2.0	Fully survived	None	5/4	4/3	6

F, female; M, male; Rt., right; Lt., left; 2-PDT, two-point discrimination test.

**Table 2 jcm-11-03394-t002:** Summary of the two-point discrimination test data.

	Static 2-PDT (Affected Fingertip)	Static 2-PDT (Contralateral Normal Fingertip)	Dynamic 2-PDT (Affected Fingertip)	Dynamic 2-PDT (Contralateral Normal Fingertip)
Good (Static 2-PDT, <6 mm; dynamic 2-PDT, <4 mm)	13	15	8	13
Fair (Static 2-PDT, 6~10 mm; dynamic 2-PDT, 4~7 mm)	2	0	7	2
Poor (Static 2-PDT, >10 mm; dynamic 2-PDT, >7 mm)	0	0	0	0
Mean ± SD (mm)	4.00 ± 1.13	3.60 ± 0.63	3.27 ± 1.03	2.87 ± 0.64
Student’s *t*-test	*p* = 0.243		*p* = 0.213	

2-PDT, two-point discrimination test; SD, standard deviation.

## Data Availability

The data presented in this study are available on request from the corresponding author. The data are not publicly available due to privacy restrictions.

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
