# Peer review of "Modified Mini-Keystone Flaps for Coverage of Tiny Volar Pulp Defects of the Fingertips in Cases with Missing Amputation Skin Stumps: A Retrospective Study"

_jcm, 2022, doi:10.3390/jcm11123394_

Round 1

Reviewer 1 Report

The basic principle of keystone flap is that the flap edges should be divided to be called a keystone flap , what we see in this paper is a double transposition flaps not a keystone flap.

I think this paper title should be changed because this flap is not a keystone flap

Author Response

Response to Reviewer 1

The basic principle of keystone flap is that the flap edges should be divided to be called a keystone flap, what we see in this paper is a double transposition flaps not a keystone flap.

I think this paper title should be changed because this flap is not a keystone flap

Response: We would like to thank Reviewer 1 for the time and effort in reviewing our manuscript and providing comments and suggestions, which have considerably helped us improve our manuscript. We have answered each of your points below and hope that our responses and revisions address all your comments.

As you mentioned, the modified mini-keystone flap (m-KF) in our study differs from the original keystone flap (keystone perforator island flap) by Behan in the following point. Our modified m-KF had two modifications of the keystone flap technique, which included omega variant (OV) closure and Sydney Melanoma Unit modification (SMUM). Particularly, the defects in our study were closed in a fish-mouth fashion (OV closure), and the flaps had a skin bridge along the greater arc of the KF (SMUM). Therefore, we have used the term “the modified mini-keystone flaps (m-KF)” in the title.

Revised title: “Modified mini-keystone flaps for coverage of tiny volar pulp defects of the fingertips in cases with missing amputation skin stumps: A retrospective study”

Reviewer 2 Report

The authors reported their case series in which a modified keystone flap (a combination of the omega variation keystone flap and Sydney melanoma unit modification keystone flap) was used for pulp reconstruction. 

The introduction should be more concise. The authors should refer to the keystone flap reconstruction and its modification briefly in the intro. Besides, I don’t think most readers know what the ‘modified mini-keystone flap’ is.

Was 2-PD measured at the original tip or the KF flap?
Was the bone exposed?

The authors should have carefully considered the indication of the m-KF reconstruction. I would do conservative treatments for cases 5 & 6 in which the defect was small and superficial.

Author Response

Response to Reviewer 2

The authors reported their case series in which a modified keystone flap (a combination of the omega variation keystone flap and Sydney melanoma unit modification keystone flap) was used for pulp reconstruction.

The introduction should be more concise. The authors should refer to the keystone flap reconstruction and its modification briefly in the intro. Besides, I don’t think most readers know what the ‘modified mini-keystone flap’ is.

Response: We would like to thank Reviewer 2 for the time and effort in reviewing our manuscript and providing comments and suggestions, which have considerably helped us improve our manuscript. We have answered each of your points below and hope that our responses and revisions address all your comments.

According to your advice, we have revised the Introduction section as follows (pages 1-2; lines 40-47):

“Surgeons usually choose an appropriate reconstructive method on a case-by-case basis according to the location and size of each defect [4]. Although various flap techniques have been devised to cover fingertip defects, new or modified techniques can yield favorable or better outcomes compared with previous methods. Herein, we present a retrospective review of our clinical experience with tiny volar pulp defect coverage using a modified mini-keystone flap (m-KF), that is, a combination of omega variation (OV) closure and Sydney Melanoma Unit modification (SMUM). Through this study, we demonstrate the expanding versatility of KF reconstruction in fingertips.”

Was 2-PD measured at the original tip or the KF flap?

Response: According to your advice, we have described the related contents in the Evaluation of postoperative functional outcomes sub-section as follows (page 3; lines 87-89):

“All tests were performed and recorded for the flap coverage area of the affected fingertip and the equivalent area of the contralateral normal fingertip by our senior author.”

Was the bone exposed?

Response: According to your advice, we have described the related contents in the Results section as follows (page 3; lines 107-108):

“All defects corresponded to the aforementioned definition of a tiny full-thickness defect with no bone exposure.”

The authors should have carefully considered the indication of the m-KF reconstruction. I would do conservative treatments for cases 5 & 6 in which the defect was small and superficial.

Response: As you mentioned, small and superficial fingertip defects can be managed with conservative treatment through secondary intention. However, the conservative treatment of fingertip defects has some disadvantages. We have described the related contents in the Discussion section as follows (page 9; lines 225-232):

“Secondary intention with conservative dressing, such as occlusive or semi-occlusive dressing, may actually represent the simplest solution for any small and superficial fingertip defects that do not involve the bone. However, it is usually a time-consuming treatment method and is often compromised by delayed wound healing due to an underlying disease or concurrent wound infection. Furthermore, patients may complain of persistent pain until epithelialization progresses to some extent. Through fingertip defect coverage with skin graft and local flap, surgeons can perform more time-saving treatment methods and achieve reliable wound healing for patients.”

Reviewer 3 Report

Dear Authors,

Thank you very much for submitting your interesting paper on the use of modified keystone flap for fingertip reconstruction.

The paper is well written and your results are encouraging.

Please find below my comments.

Could you please better describe the wounds you were treating? Any bone exposure? Full thickness?

What was the time to healing in your case series?

Case 1 and case 2 look like cases that could heal by secondary intention in about 2 to 3 weeks, could you please highlight the benefit of doing a surgical procedure with additional scars compared to secondary intention wound healing?

Could you please add in the “discussion” a paragraph on flap design? Scars in the pulp area could be painful and annoying for the patients. Careful planning of where the scars will be, in my opinion, is key to a good reconstruction.

Free flaps are a viable options for fingertip reconstruction, could you mention that in “discussion”?

Line 134 Tension-free flap insertion with donor-site closure was achieved

Line 149 Tension-free flap insertion with donor-site closure was achieved

Looking at the pictures it doesn’t look like you achieved a tension-free inset and closure. Could you please modify the sentences with minimal tension as you have stated in “discussion”

Thank you.

Author Response

Response to Reviewer 3

Dear Authors,

Thank you very much for submitting your interesting paper on the use of modified keystone flap for fingertip reconstruction.

The paper is well written, and your results are encouraging.

Please find below my comments.

Could you please better describe the wounds you were treating? Any bone exposure? Full thickness?

Response: We would like to thank Reviewer 3 for the time and effort in reviewing our manuscript and providing comments and suggestions, which have considerably helped us improve our manuscript. We have answered each of your points below and hope that our responses and revisions address all your comments.

According to your advice, we have described the related contents in the Results section as follows (page 3; lines 107-108):

“All defects corresponded to the aforementioned definition of a tiny full-thickness defect with no bone exposure.”

What was the time to healing in your case series?

Response: According to your advice, we have described the related contents in the Results section as follows (page 3; lines 114-115):

“All other patients were completely healed without any wound complications at 2-week postoperative follow-up.”

Case 1 and case 2 look like cases that could heal by secondary intention in about 2 to 3 weeks, could you please highlight the benefit of doing a surgical procedure with additional scars compared to secondary intention wound healing?

Response: According to your advice, we have described the related contents in the Discussion section as follows (page 9; lines 225-232):

“Secondary intention with conservative dressing, such as occlusive or semi-occlusive dressing, may actually represent the simplest solution for any small and superficial fingertip defects that do not involve the bone. However, it is usually a time-consuming treatment method and is often compromised by delayed wound healing due to an underlying disease or concurrent wound infection. Furthermore, patients may complain of persistent pain until epithelialization progresses to some extent. Through fingertip defect coverage with skin graft and local flap, surgeons can perform more time-saving treatment methods and achieve reliable wound healing for patients.”

Could you please add in the “discussion” a paragraph on flap design? Scars in the pulp area could be painful and annoying for the patients. Careful planning of where the scars will be, in my opinion, is key to a good reconstruction.

Response: According to your advice, we have described the related contents in the Discussion section as follows (pages 10-11; lines 278-283):

“Finally, any scars in the volar pulp of fingertip could be painful and annoying for patients; therefore, surgical procedures, such as our modified m-KF, should be performed carefully. As mentioned earlier, we designed the flap at the side of sufficient tissue laxity (Figure 1). The above-mentioned tension-reducing effect of the modified m-KF can allow wound closure with minimal tension, possibly helping in minimizing scar formation in the volar pulp area.”

Free flaps are viable options for fingertip reconstruction, could you mention that in “discussion”?

Response: According to your advice, we have described the related contents in the Discussion section as follows (page 8; lines 184-186):

“Considering these objectives, free flaps are suitable for large fingertip reconstruction, whereas local flaps can be the best choice for small-to-moderate-sized defect coverage of the fingertip [1,2,11].”

Line 134 Tension-free flap insertion with donor-site closure was achieved

Line 149 Tension-free flap insertion with donor-site closure was achieved

Looking at the pictures it doesn’t look like you achieved a tension-free inset and closure. Could you please modify the sentences with minimal tension as you have stated in “discussion”

Response: According to your advice, we have revised the sentences as follows (lines 136, 152, and 167):

“Flap insertion and donor-site closure with minimal tension were achieved.”

Round 2

Reviewer 2 Report

I still consider indications were overestimated in some cases, and the aesthetic result was not satisfactory: the contour of the fingertip was not roundish but concave in Figure 4. Limited aesthetic results should be described as one limitation of the procedure in the discussion. However, the authors answered my concern. I hope this article helps the readers treat fingertip wounds.

Author Response

Response to Reviewer 2 comment I still consider indications were overestimated in some cases, and the aesthetic result was not satisfactory: the contour of the fingertip was not roundish but concave in Figure 4. Limited aesthetic results should be described as one limitation of the procedure in the discussion. However, the authors answered my concern. I hope this article helps the readers treat fingertip wounds.

Response: Thank you for your valuable comment. According to your suggestion, we have revised the Discussion section as follows (pages 10; lines 278-281): Finally, any scar at the volar pulp of the fingertip can be painful and irritating for patients. Limited aesthetic results, such as change in fingertip contour, can occur due to surgical reconstructions; therefore, surgical procedures such as our modified m-KF should be performed with caution.
